# Effectiveness of Deep Water Running on Improving Cardiorespiratory Fitness, Physical Function and Quality of Life: A Systematic Review

**DOI:** 10.3390/ijerph19159434

**Published:** 2022-08-01

**Authors:** Manny M. Y. Kwok, Billy C. L. So, Sophie Heywood, Matthew C. Y. Lai, Shamay S. M. Ng

**Affiliations:** 1Gait and Motion Analysis Laboratory, Department of Rehabilitation Sciences, The Hong Kong Polytechnic University, Hong Kong, China; manny.kwok@connect.polyu.hk (M.M.Y.K.); cyiulai@polyu.edu.hk (M.C.Y.L.); shamay.ng@polyu.edu.hk (S.S.M.N.); 2Physiotherapy Department, St Vincent’s Hospital Melbourne, Fitzroy, VIC 3065, Australia; sophie.heywood@svha.org.au; 3Melbourne Sports Medicine Centre, Melbourne, VIC 3000, Australia; 4Hydro Functional Fitness, Melbourne, VIC 3000, Australia

**Keywords:** aquatic exercise, conditioning, sport, physical fitness, wellness

## Abstract

Deep Water Running (DWR) is a form of aquatic aerobic exercise simulating the running patterns adopted on dry land. Little is known on the effectiveness of DWR despite gaining popularity. The objective of this study is to systematically review the effects of DWR on cardiorespiratory fitness, physical function, and quality of life in healthy and clinical populations. A systematic search was completed using six databases, including SPORTDiscus, MEDLINE, CINAHL, AMED, Embase, and The Cochrane Library, up to February 2022. Eleven studies evaluating the effectiveness of DWR on cardiorespiratory fitness (CRF), physical function, or quality of life (QoL), compared with no interventions (or standard treatment) or land-based trainings were identified. Data relevant to the review questions were extracted by two independent reviewers when means and standard deviations were reported, and standardized mean differences were calculated. A quality assessment was conducted using selected items from the Downs and Black checklist. A total of 11 clinical trials (7 randomized controlled trials) with a total of 287 participants were included. Meta-analyses were not completed due to the high level of clinical and statistical heterogeneity between studies. Compared with land-based training, DWR showed similar effects on CRF with limited studies reporting outcomes of physical function and QoL compared with the no-exercise control group. DWR appears to be comparable to land-based training for improving CRF. The aquatic environment may provide some advantages for off-loaded exercise at high intensity in populations that are weak, injured or in pain, but more studies are required.

## 1. Introduction

Deep water running (DWR) has gained popularity among different patient populations in recent decades for cardiorespiratory conditioning. DWR can be defined as running with 70% of the body immersed, submerged at shoulder level, with or without the use of a floatation device [1]. It simulates land running movement patterns performed but is conducted in a non-weight-bearing aquatic environment. With the advantage of the unloading property of buoyancy, DWR eliminates vertical ground reaction forces, and hence reduces joint loadings and the potential risk for injury to the musculoskeletal system [2,3].

Changes in the properties of water, including temperature, as well as the instructions for training, can influence the outcomes of DWR. For instance, water temperature can directly influence the physiological mechanisms brought about by DWR. A temperature of 26 °C to 28 °C is recommended for DWR performed at maximal intensities [1]. This can result in a reduction in maximal heart rate (HRmax), increased energy expenditure, and body heat dissipation as compared with land control groups [4]. Speed of movement can also influence drag force, and the level of resistance overload depends on the speed of the movement [5]. At a given speed, when the surface area of the limb is increased, an additional overload can be achieved [6]. Therefore, a combination of changes in water temperature and drag force can promote DWR program progression [3].

Deep water running has emerged as an alternate exercise for athletic, healthy untrained populations as well as people with musculoskeletal conditions [1,7]. During DWR, the buoyancy unloading effect of the water negates the weight-bearing nature of exercise relative to that conducted on land. The lower impact force experienced in the water allows the exercise to be performed both at higher intensities and for longer durations, with a potential reduction in risk of injury [8,9]. Although DWR may induce lower physiological responses when compared with land running with a lower maximal oxygen uptake (VO_2_ max) and HRmax [1], the training stimulus is likely to still be sufficient for supplementary training, with decreased orthopaedic trauma compared to land running [1,10]. As such, the reduction in loading on the body makes DWR a viable training alternative for athletes who are susceptible to repetitive stress injuries, as well as for elderly individuals who may be vulnerable to, or fearful of, falls or joint pain.

Although it is known that DWR provides a form of unloaded, non-weight-bearing exercise, the biomechanics of DWR when compared with land running remains unclear. DWR can be classified into two styles, namely, (1) the cross-country style, and (2) the high knee style [11]. While deep water running instructions on style may vary, the consensus for techniques follow these principles: (1) the trunk should be slightly forward in upright position, (2) the arms and legs should follow a circular leg and arm motion, (3) the swing phase leg should be brought to the horizontal position, (4) the elbows should be maintained at a 90° angle, and (5) the hands should remain either completely open or in a closed fist at all times [12]. There may not be consensus on whether the DWR style should be attempting to replicate similar lower limb kinematics, or whether larger ranges of movement lead to greater resistance and higher intensity. A recent review suggested that muscle activity during DWR demonstrated a higher activation of distal leg muscles than proximal thigh muscles, in comparison with land treadmill running [3], although it is not clear what physical functional outcomes this may influence. Hence, there is increasing interest in examining the differences in physical functional outcomes with validated measurement tools for DWR and land running.

The effect of DWR on health-related quality of life (QoL) remains unclear. It is common for people with musculoskeletal conditions to report lower levels of QoL—including in domains related to physical function—due to fear and pain [13]. The fear of physical activity, arising from the feeling of pain during movement or risk of reinjury, often leads to a downward spiral of negative physical and psychological consequence [14]. Deep water running, a proposed low impact aerobic exercise, may offer a potential solution to mitigate fear associated with pain.

Despite the fact that there is growing interest in the inclusion of DWR in cardiovascular training programs for various clinical populations, and a number of clinical trials have been conducted to evaluate the effects of DWR on numerous quality-of-life domains, there has not yet been a systematic examination of the effects of DWR on CRF, physical function, and QoL. The current systematic review was conducted to address this gap. Specifically, we sought to evaluate the effects of deep water running (DWR) on CRF, physical function, and QoL, when compared to other interventions or control groups without exercise.

## 2. Method

### 2.1. Study Selection

Two reviewers (MK and BS) screened the article titles and abstracts. The reviewers independently reviewed the full texts and reached an agreement for the eligibility to be included in the review. If in disagreement, a third reviewer (SH) resolved any discrepancies between the two reviewers, and the article was removed from the review.

### 2.2. Search Strategy

This systematic review was guided by the Preferred Reporting Items for Systematic reviews and Meta-analysis (PRISMA) guidelines [15], and was registered in PROSPERO database (CRD42020154988) prior to conducting the review. A complete search of six databases: SPORTDiscus, MEDLINE, CINAHL, AMED, Embase, and The Cochrane Library, were conducted up to February 2022 with a combination of text words listed in Appendix A. In an effort to unveil complex evidence in obscure locations, manual searches of “reference tracking” and “citation tracking” of relevant articles were completed [16]. With the use of citation tracking databases (i.e., “Cited by” in Google Scholar), the forward tracking of selected studies was processed.

### 2.3. Inclusion Criteria

The articles included were based on the following criteria: (1) Studies must be interventional studies, either randomised or non-randomised trials; (2) studies must be published as a full paper in English; (3) study participants were not limited to only healthy individuals (i.e., included patients with at least one health condition); (4) the interventions being evaluated were predominantly DWR (≥50%)-based or combined with passive treatments (e.g., education or stretching); and (5) outcomes included cardiorespiratory measures (e.g., VO_2_ max, maximal heart rate, blood pressure, ventilatory threshold), physical function related to running or walking (e.g., timed test, endurance test, or functional test), or health-related QoL (validated questionnaires or assessment tools) (Appendix A).

### 2.4. Exclusion Criteria

The studies were excluded if: (1) The study designs were non-experimental (e.g., descriptive or exploratory studies) and/or cross-sectional studies; (2) participants were under 18 years of age; (3) they studied exercises other than DWR (e.g., general aquatic exercise, shallow water running, underwater treadmill) (Appendix A).

### 2.5. Data Extraction

Data were extracted independently by 2 reviewers (MK and BS) and included study design, baseline demographics of study participants (age, sex, health status, body weight, height, body mass index), sample size, intervention details (frequency, intensity, duration, length of intervention, DWR technique, instructions, supervision of exercise intervention, equipment used, pool temperature, and room temperature), outcomes of interest (measurements on cardiorespiratory outcomes, physical functional outcomes and health-related QoL), and measures of variance of the outcomes of interest. If there were missing data or data potentially included in error, the study authors were contacted via email and asked to provide further information.

### 2.6. Quality and Risk of Bias Assessment

Each included study was critically appraised by two independent reviewers (MK and BS) using selected items from a checklist based on Downs and Black (Table 1) [17]. In the preliminary search of articles, relevant studies of multiple study designs were drawn. The Downs and Black quality assessment tool designed for randomized and non-randomized trials was chosen to assess the quality. Despite there being limitations, it has been shown to be an effective tool irrespective of the tool development and extensive domains covered [18]. Five subscales, including reporting, external validity, internal validity in bias, confounding, and power were assessed according to the characteristics of reviewed studies. Scores for this tool can be interpreted as follows: Excellent (26–28); good (20–25); fair (15–19); and poor (≤14) [17].

### 2.7. Data Analysis

The outcomes of interest and program parameters in specific populations were pooled for data analysis using Cochrane Software Review Manager (RevMan software, version 7.1; The Nordic Cochrane Centre, Copenhagen, Denmark). The effects of DWR on the three domains of interest were analysed with standardized mean difference (SMD) calculations if the studies reported means and standard deviations [29]. The SMD was significant only if the 95% CI did not include zero [30]. Effect-size thresholds were classified as small effect (0.2), medium effect (0.5), or large effect (0.8) [29]. The SMD and 95% confidence intervals (CI) were reported to demonstrate effects of DWR in comparison with a control group (without exercise or other interventions such as land-based training). Meta-analyses were not completed due to Downs and Black poor-quality score of (≤14) [10,19,20,25,26,27,28]. Therefore, results are presented as effect sizes only.

## 3. Results

### 3.1. Selection of Studies

Following an initial search identification of 1416 articles, 11 articles were included in the review (Figure 1) for quantitative synthesis.

### 3.2. Study Quality

Of the eleven included studies, seven studies were randomized controlled trials (RCT)s [10,19,20,21,22,23,24,25,26,27,28], two were longitudinal studies [19,21], and two were quasi-experiment studies [25,27].

Four studies investigated physically active or trained adults [20,26,27,28], three studies investigated physically inactive or obese women [10,19,21], two studies investigated healthy or community-dwelling elderly [10,24], and one study investigated populations with lower-back pain [20]. Study participant numbers varied from 10 to 60, with a total of 287 participants in the review (Table 2).

As for the comparison group, five studies included a control group that did not participate in any exercise [10,21,23,24,25], and six studies included a land-based exercise group [19,20,22,26,27,28]. Among them, two studies matched land-based exercise time to DWR, having an identical total of 45 sessions over a three-week period [22,23]. The rest of the studies varied in exercise intensity and number of sessions [10,19,20,21,24,25,26,27,28].

### 3.3. Deep Water Running Intervention Characteristics

The duration of DWR interventions varied from 30 to 70 min, 2 to 5 sessions per week, lasting between 3 to 18 weeks, with a compliance rate range from 80% to 100% (Table 3). Of the 11 studies, 10 studies had participants performing DWR with the assistance of flotation devices [10,19,20,21,22,23,24,25,26,27,28]. Only in the study of McKenzie and Mcluckie (1991), participants were asked to not touch the bottom of the swimming pool without wearing flotation devices during DWR [26].

All the studies instructed participants to perform running in the water; however, there were variations in body positions during DWR. Four studies required participants to maintain a vertical body position during DWR [20,22,26,28]. Davidson and McNaughton (2000) allowed participants to perform DWR in a vertical or slightly forward leaning position, while Broman et al. (2006) and Michaud et al. (1995) instructed participants to run in a slightly forward bending position [10,19,25]. Three studies did not specify the running positions adopted [21,23,24,27].

The included papers’ analyses revolved the varied protocol types measuring cardiorespiratory fitness. Maximal oxygen uptakes were measured by an incremental exercise test on a treadmill [19,20,25,26,27,28] or cycle ergometer [10,28]. Most papers did not report any pretest screening, with some studies using health questionnaires [10,19,20,25,26,27,28]. Familiarisation sessions before testing were mentioned by most authors, and more specific details were provided in seven of the studies [10,19,20,25,26,27,28]. Indications for test termination were described in most studies; however, they often lacked objective absolute indications. Incremental exercise protocol end points mainly occurred when participants arrived at volitional exhaustion [10,19,20,25,26,27,28].

Of the 11 articles, all studies clearly described the objectives and main outcomes, which were reliable and valid. Nine out of eleven studies failed to report concealment of allocation [10,19,21,22,24,25,26,27,28]. Only two studies reported blinding the outcome assessors or intention-to-treat analysis [20,22].

### 3.4. Outcomes

Among the eleven included studies, three studies investigated physical functional outcomes (functional mobility and balance tests) or walking [24,26,28], nine studies investigated CRF [10,19,20,21,22,25,26,27,28], and three studies investigated QoL [20,22,23].

#### 3.4.1. Cardiorespiratory Fitness

Peak Heart rate (HR peak)

Three studies evaluated the changes in HR peak [10,25,27], but did not report means and SD; therefore, SMDs could not be calculated.

Maximal Aerobic Capacity (VO_2_ max)

Deep Water Running Versus Control (no active exercise)

One study, Broman et al. (2006), found a significant large effect (SMD = −1.28) of DWR on improving VO_2_ max [10]. In contrast, the other study, Michaud et al. (1995), observed no significant difference in effect of DWR for improving VO_2_ max when compared to no exercise (SMD = −0.83) [25] (Figure 2).

Deep Water Running Versus Land Training

Similar effects were found by five individual studies between DWR and land-based training for improving VO_2_ max [19,20,26,27,28] (Figure 3). Among athletic populations, VO_2_ max ranged from SMD 0.06–0.10 [26,27]. For individuals with health conditions or in the untrained healthy population, VO_2_ max was −0.83 [20] and ranged from SMD −0.39–0.28, respectively [19,28].

#### 3.4.2. Physical Function

Deep Water Running Versus Control (no active exercise)

One study, Alberti et al. (2017) evaluated physical function between the DWR group and control group with three walking tests (4MWT, 6MWT, 10MWST), and two functional mobility and dynamic balance tests: Five Times Sit to Stand Test (FTSST) and Timed Up and Go Test (TUGT) [24]. Large effect sizes of DWR were observed in the three walking tests (SMD = −2.39 to −1.33). Effect sizes revealed no significant effect of DWR compared to the no active exercise for FTSST and TUGT (SMD = −0.49 to −0.36), although there were mean improvements favouring the DWR group in this study (Figure 4).

Deep Water Running Versus Land Training

Two studies measured running outcomes, reporting similar effects on physical function between DWR and land training groups (Figure 5). Mckenzie et al. (1991) found a similar effect on the time to reach volitional exhaustion (i.e., time to fatigue) while running on a treadmill in DWR and land running groups (SMD = 0.82) [26]. Eyestone et al. (1993) found a similar effect of DWR to both land-running and cycling for two-mile run time (SMD = −0.41 to −0.27) [28].

#### 3.4.3. Quality of Life

Deep Water Running Versus Control (no active exercise)

One study measuring QoL, Cuesta-Vargas et al. (2012), found a large effect of DWR (SMD = −2.01 to −3.23) in both mental and physical subsets of data in improving QoL compared with the no-exercise control group [23] (Figure 6).

Deep Water Running Versus Land Training

Two studies found similar effects of DWR and land training on improving QoL (SMD = 0.03 to 0.05) [20,22] (Figure 7).

## 4. Discussion

This systematic review found similar effects of DWR and land-training on improving VO_2_ max in active individuals, sedentary participants and people with chronic health conditions. Similar effects were also found for physical function and QoL outcome measures for water-based and land-based training; however, the conclusions for these domains are weaker due to the smaller number of studies. There were few studies reporting the effect of DWR to no-exercise across all three domains of interest with mixed results. Heterogeneity and reporting of interventions, as well as the varied types of participants, limits stronger conclusions.

The results of the review reflect that DWR can be an equally effective form of exercise to maintain or improve CRF when compared with land-based training. VO_2_ max is considered to be the highest value of VO_2_ attained upon an exercise stress test [31]. The test of VO_2_ max is designed to bring the subject to the limit of exhaustion, a gold-standard measure of CRF [32]. VO_2_ max can be influenced by a restriction in chest expansion when inspiratory muscle contractions are unable to equal or overcome the force of hydrostatic pressure under immersion [33]. It is suggested that central hypervolaemia, or the shift of blood into the chest cavity, reduces lung volume, as well as reducing lung compliance and promoting gas trapping, which narrows the airways [5,34]. Compression of both the abdomen (in turn pushing up the diaphragm) and the chest wall itself by water also increases the work of breathing [5,35]. With such unique physiological adaptations during aquatic training, minute ventilation and breathing frequency could potentially be increased when compared with land-based training with an equivalent exercise intensity [36]. Additionally, it is suggested that by performing movements in DWR, the physiological effects of water immersion contribute to a reduction in joint loading, while the tactile, thermal stimulation and drag force may enhance joint proprioception, body balance and muscle strength [22]. Since aquatic-based maintenance of cardiorespiratory conditioning offers a number of additional advantages over land-based training, it may be ideal for populations unable to exercise on land, or those who exclusively train on land, and desire to cross-train in water for rehabilitative purposes.

This review found that DWR is more effective in untrained, sedentary healthy elderly populations [10,25] than trained or physically active subjects [19,20,26,27,28]. One possible explanation of DWR improving aerobic capacity in sedentary individuals and elderly populations is their low initial fitness [1]. Of note, such a substantial improvement likely has more significant clinical relevance in sedentary healthy elderly populations than trained or physical active subjects, given that cardiorespiratory function decreases with primary ageing, and that CRF declines steadily in sedentary individuals at a rate of approximately 1% per year after the third decade of life [37]. Such findings are in agreement with a previous study by Reilly et al. (2003) [1], and other systematic reviews by Chu and Rhodes (2001) [12] and Jorgic et al. (2012) [38]. As such, CRF training, for instance DWR, promotes improvement in CRF and has shown important clinical implications for sedentary populations, particularly in individuals with advanced age.

A key component of aquatic-based training is carryover to land function as well as broader QoL improvements. One of the aims of this review was to analyse the effect of DWR compared to land-based walking and running. This review found a large effect for DWR in walking tests (4MWT, 6MWT and 10MWT) when compared with the no-active-exercise control groups, but only one study reported this [24]. Similarly, few data on the effect of DWR on quality were found. When DWR was compared with the no-active-exercise control group, DWR showed a significant effect upon physical health and mental health in SF-12, but only in one study [23]. There is a potential for QoL to be influenced via changes in physical function or to improve well-being by promoting relaxation, vasodilation, and a reduction in joint loading, and to produce an analgesic effect [22]. DWR may also release cortisol and adrenaline into the bloodstream, thus increasing the pain threshold for those subjects who suffered from pain [23], along with the removal of metabolic waste and reduction in nociceptor activation [39]. More evidence is required to understand if DWR has enough of an effect to improve both land-based function and QoL.

### 4.1. Study Limitations

Although the synthesized evidence in this review is encouraging, this study has several limitations. Firstly, the majority of the included studies had small sample sizes, and the recruitment of participants was from convenience sampling. This may have increased the chance of committing a type II error, affecting the results of efficacy regarding subjects’ representativeness and generalisability. Secondly, heterogeneity of participants’ characteristics also made it difficult to draw definitive conclusions about clinical outcomes. Furthermore, a meta-analysis was not completed due to the clinical diversity in study interventions, variations in methodology, and risk of bias in individual studies. For instance, there were a wide range of DWR protocols and methods of measurements for the main outcomes; that is, the measurements used or methods in measuring VO_2_ max were distinct across some studies. This limits the ability of this review to conclude a recommended dosage of DWR for participants. As a result, a limited number of studies on homogeneous groups of subjects have been conducted in this review. Lastly, the lack of a meta-analysis hinders a precise estimation of effect and a statistical analysis of DWR. More clinical trials in specific populations with larger sample sizes could help yield more solid conclusions and consistency on the effectiveness of DWR.

### 4.2. Future Directions

Conducting correlational studies could be useful with the application of biopsychosocial approaches, to investigate the relationships between measured outcomes and DWR with a larger and more homogenous group of participants. This could further justify the clinical significance of DWR in specific groups of populations. Additionally, the studies measured short-term effects immediately after DWR sessions. However, long-term effects could be considered after completion of intervention to evaluate the carryover effects. Currently, there is a lack of evidence of participants’ experience and perceptions towards DWR programs. Qualitative research is therefore suggested to explore the attitudes, behaviours, beliefs, and satisfaction towards effects of DWR to further consider their exercise adherence in addition to quantitative data on both land-based function and QoL. It is also difficult to conclude optimal dosages of a training programme; therefore, future investigations on the programme intensity are warranted.

## 5. Conclusions

DWR appears to be comparable to land-based training, with mixed results of effects compared to no exercise for improving CRF, physical functions and QoL. The small number of studies and quality of the evidence limits further conclusions. To further understanding of the potential benefits of DWR, future research is needed to develop an effective prescription for targeted populations.

## Figures and Tables

**Figure 1 ijerph-19-09434-f001:**
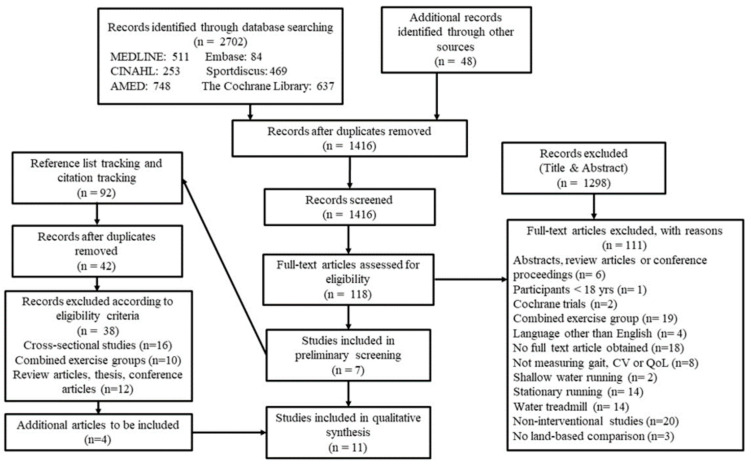
PRISMA flow diagram of selection process; n = number of publications.

**Figure 2 ijerph-19-09434-f002:**
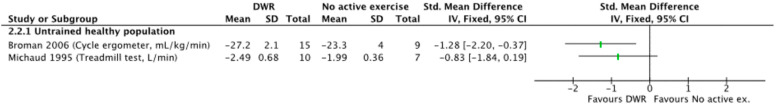
Standardized mean difference (95% CI) for the effect of DWR compared with no active exercise on VO_2_ max [10,25].

**Figure 3 ijerph-19-09434-f003:**
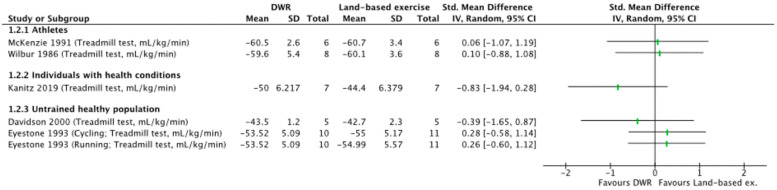
Standardized mean difference (95% CI) for the effect of DWR compared with land trainings on VO_2_ max [19,20,26,27,28].

**Figure 4 ijerph-19-09434-f004:**
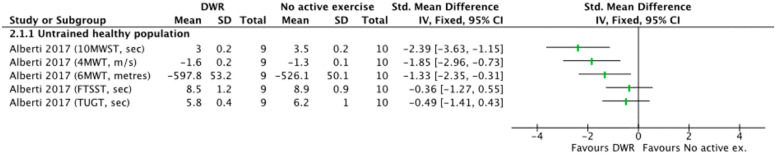
Standardized mean difference (95% CI) for the effect of DWR compared with no active exercise on physical function outcomes [24].

**Figure 5 ijerph-19-09434-f005:**
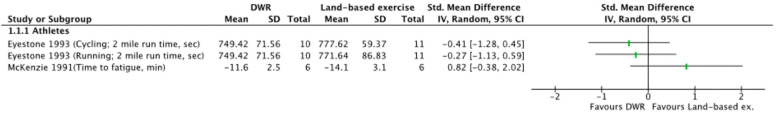
Standardized mean difference (95% CI) for the effect of DWR compared with land trainings on physical function outcomes [26,28].

**Figure 6 ijerph-19-09434-f006:**
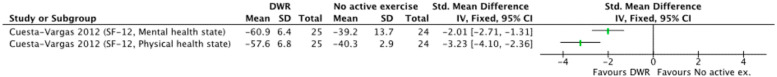
Standardized mean difference (95% CI) for the effect of DWR compared with no active exercise on QoL [23].

**Figure 7 ijerph-19-09434-f007:**
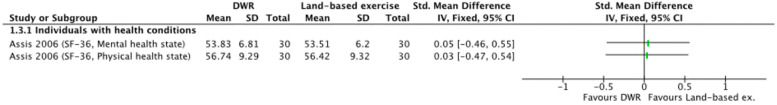
Standardized mean difference (95% CI) for the effect of DWR compared with land trainings on QoL [22].

**Table 1 ijerph-19-09434-t001:** Quality Assessment.

Subscale	Items	Davidson, K., and McNaughton, L (2000) [19]	Kanitz et al. (2019) [20]	Colato et al. (2016) [21]	Assis et al. (2006) [22]	Broman et al. (2006) [10]	Cuesta-Vargas et al. (2012) [23]	Alberti et al. (2017) [24]	Michaud et al. (1995) [25]	Mckenzie et al. (1991) [26]	Wilber et al. (1996) [27]	Eyestone et al. (1993) [28]
Reporting	1. Hypothesis/aim/objective clearly described	Y	Y	Y	Y	Y	Y	Y	Y	Y	Y	Y
	2. Main outcomes clearly described	Y	Y	Y	Y	Y	Y	Y	Y	Y	Y	Y
	3. Characteristics of the patients clearly described	N	Y	Y	Y	Y	Y	Y	N	N	Y	N
	4. Intervention and comparison group clearly described	Y	Y	N	Y	N	N	N	Y	Y	Y	Y
	5. Distributions of principal confounders in each group of subjects clearly described	N	Y	Y	Y	Y	Y	Y	N	Y	N	N
	6. Main findings clearly described	Y	Y	Y	Y	Y	Y	Y	N	Y	Y	Y
	7. Estimates of the random variability for the main outcomes provided	N	Y	Y	Y	Y	Y	Y	N	Y	Y	Y
	10. Actual p values reported for main outcomes	Y	N	Y	Y	N	Y	Y	N	N	N	Y
External validity	11. Subjects asked to participate represented the population	N	N	Y	Y	Y	Y	N	N	N	N	N
	12. Subjects prepared to participate represented the population	N	N	N	N	N	N	N	N	N	N	N
Internal validity-Bias	15. Blinded outcome assessment	N	Y	N	Y	N	N	N	N	N	N	N
	18. Appropriate statistical tests performed	Y	Y	Y	Y	Y	Y	Y	N	N	Y	Y
	20. Accurate outcome measure used (reliable and valid)	Y	Y	Y	Y	Y	Y	Y	Y	Y	Y	Y
Internal validity-Confounding	23. Subjects randomized to intervention groups	N	Y	N	Y	Y	Y	N	N	Y	N	Y
	24. Concealed allocation from subjects and investigators	N	Y	N	N	N	Y	N	N	N	N	N
	26. Losses to follow-up taken into account	Y	Y	Y	Y	Y	Y	Y	Y	Y	Y	N
Power	27. Power calculation	N	Y	Y	Y	N	Y	Y	N	N	N	N

**Table 2 ijerph-19-09434-t002:** Study design of included studies.

	Number of Subjects (M/F)	Age (Years Old) of Respective Groups	Subjects’ Characteristics	Outcome Measure of Physical Functions	Outcome Measure of CRF	Outcome Measure of QoL	Study Design
Alberti et al. (2017) [24]	19 (0/19)	DWR: 64.33 ± 4.24 Control: 64.40 ± 4.22	Community dwelling elderly	4MWT, 6MWT, 10MWST, FTSST TUGT	/	/	R
Assis et al. (2006) [22]	60 (0/60)	DWR: 43.96 ± 10.28 LBE: 44.04 ± 8.87	Sedentary women with fibromyalgia	/	HR, VO_2_max	SF-36	R
Broman et al. (2006) [10]	29 (0/29)	DWR: 69.0 ± 4.0 Control: 69.8 ± 3.5	Healthy elderly women	/	HR (rest test) BP (rest test) Peak VO2	/	R
Colato et al. (2016) [21]	20 (0/20)	DWR:48.81 ± 12.87 Control: 49.9 ± 10.5	Overweight obese women	/	VO_2_max	/	L
Cuesta-Vargas et al. (2012) [23]	58 (25/33)	DWR: 38.6 ± 12.2 Control: 37.8 ± 13.2	Non-specific low back pain	/	/	SF-12	R
Davidson, K., & McNaughton, L. (2000) [19]	10 (0/10)	22.6 ± 3.4	Untrained women	/	VO_2_max	/	L
Eyestone et al. (1993) [28]	32 (32/0)	18–26	Finished a 1.5 mile run in less than 10’45	2 mile run time	VO_2_max	/	R
Kanitz et al. (2019) [20]	14 (7/7)	DWR: 39 (95% CI: 31–47) LWR: 40 (95% CI: 36–50)	Physically active patients of both sexes with chronic low back pain	/	VO_2_peak, VO_2_, Vt_2_	/	R
Mckenzie et al. (1991) [26]	12 (12/0)	23.9	Competitive runners	Time to fatigue	VO_2_max	/	R
Michaud, T. J. et al. (1995) [25]	17 (2/15)	DWR: 32.6 ± 6.8 Control: N/A	Healthy sedentary	/	VO_2_max	/	Q
Wilber et al (1996) [27]	16 (16/0)	32.5 ± 5.4	Aerobically trained male distance runners	/	VO_2_max, Ventilatory threshold	/	Q

Note: 4MWT: 4-meter walk test; 6MWT: 6-meter walk test; 10MWST: 10-meter walking speed test; FTSST: 5 times sit to stand test; TUGT: Timed up and go test; L: Longitudinal; Q: Quasi-experimental; R: Randomized Controlled Trial.

**Table 3 ijerph-19-09434-t003:** Intervention design of included studies.

	Groups Numbers	Super- Vision	Adverse Effects %	Drop Outs %	Pool Temp ºC	Water Depth m	Floating Device	Compliance %	Program Time (Weeks)	Session Time (Mins)	Sessions per Week	Total Number of Sessions	Warmup/ Cooldown
Alberti et al. (2017) [24]	DWR = 16 Control = 14	√	N/A	7 4	28–30	1.35	√	N/A	18	50	2	36	√
Assis et al. (2006) [22]	DWR = 26 LBE = 26	√ 2 PT	10 16	4 4	28–31	N/A	√	100	15	60	3	45	√
Broman et al. (2006) [10]	DWR = 18 Control = 11	N/A	x	3 2	27	N/A	√	N/A	8	43	2	16	√
Colato et al. (2016) [21]	DWR = 11 Control = 9	N/A	N/A	N/A	28	1.7	√	N/A	12	70	3	36	√
Cues-ta-Vargas et al. (2012) [23]	DWR + GP = 29 GP = 29	√	N/A	3 4	N/A	2.15	√	N/A	15	30	3	45	N/A
Davidson, K., & McNaughton, L. (2000) [19]	DWR = 5 RR = 5	√	N/A	0 0	22–25	N/A	√	96 94	4	50	3	12	N/A
Eyestone et al. (1993) [28]	DWR = 10 C = 11 RR = 11	N/A	N/A	N/A	N/A	Diving pool	√	N/A	6	20–30	Week 1: 3 Week 2: 4 Week 3–6: 5	27	N/A
Kanitz et al. (2019) [20]	DWR = 7 LWR = 7	N/A	N/A	3 (30%) 3 (30%)	N/A	N/A	√	83 80	12	45	2	24	√
Mckenzie et al. (1991) [26]	DWR = 6 LWT = 6	N/A	N/A	N/A	N/A	Deep pool	x	N/A	3	30	5	15	N/A
Michaud, T. J. et al. (1995) [25]	DWR = 10 Control = 7	N/A	N/A	6	27–29	Diving pool	√	100	8	40–70	3	24	X
Wilber et al (1996) [27]	DWR = 8 TR = 8	N/A	N/A	1 1	27	N/A	√	96 98	6	30–60	5	30	N/A

Note: C: Cycling; DWR: Deep Water Running; RR: Road Running; LWR: Land Walking/Running; LBE: Land-based Exercises; GP: General Practice Consisting of Advice and Education about Exercise; TR: Treadmill Run; WT: Water Training; √: Included; x: Not Included; N/A: Not Available.

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
