# Peer review of "Effectiveness of Deep Water Running on Improving Cardiorespiratory Fitness, Physical Function and Quality of Life: A Systematic Review"

_ijerph, 2022, doi:10.3390/ijerph19159434_

Round 1
Reviewer 1 Report
This is a systematic review of the literature on the effects of DWR in cardiorespiratory fitness, physical function, and health related quality of life. In general, it is a well written manuscript with sound scientific methodology and reporting. I have some suggestion to improve the quality of the reporting:
– Please check all the manuscript, including tables and figures, to maintain consistency of terms and acronyms. For cardiorespiratory fitness, the most common acronym is CRF. You sometimes use cardiovascular or CV. For physical function, as an umbrella term, you often use functional or gait performance (appendix) for labelling which confuses readers.
– I would expect more detail on the evaluation of CRF, i.e., was a maximal CPET used? Conducted on a treadmill or a cycle-ergometer? Which protocol(s) were used? For example, peak HR alone may be a poor outcome measure to assess effects of training if not properly framed into more comprehensive analysis and critical appraisal of the primary studies. Later, on Limitations section you refer to those measures as very heterogenous (line 332) but it’s not possible for readers judge that unless they go and read the primary studies.
– Related to my previous point I would like more detail on the results of the primary studies, e.g., how much gain was obtained in VO2max or peak in study a, b, and c. How much gain was obtained in the physical performance tests or health related quality of life measures?
– I would swap the column headers and contents for the rows. I think ot would be more intuitive.
Specific comments:
Line 15: Please amend “healthy” because special and clinical populations were also included in the review
Line 17: Please amend “clinical trial” because not all included studies were trials, or in clinical populations.
Line 136: Table 3 instead of Appendix 3?
Line 155: “risk of bias in individual studies”, what was the threshold used to decide not to conduct meta-analysis? Please report.
Line 236: “from” should be removed.
Lines 243–247: I would like (and many other readers) details on how fatigue was assessed or the conditions of testing to critically appraise the findings.
Author Response
This is a systematic review of the literature on the effects of DWR in cardiorespiratory fitness, physical function, and health related quality of life. In general, it is a well written manuscript with sound scientific methodology and reporting. I have some suggestion to improve the quality of the reporting:
Response: Thank you very much for your comments. We tried our test to address your suggestions point by point. Cheers.
– Please check all the manuscript, including tables and figures, to maintain consistency of terms and acronyms. For cardiorespiratory fitness, the most common acronym is CRF. You sometimes use cardiovascular or CV. For physical function, as an umbrella term, you often use functional or gait performance (appendix) for labelling which confuses readers.
Reponse: Apologies for the inconsistency, we have checked the consistency of the terms and acronyms in the text, tables and appendix. Thank you very much.
– I would expect more detail on the evaluation of CRF, i.e., was a maximal CPET used? Conducted on a treadmill or a cycle-ergometer? Which protocol(s) were used? For example, peak HR alone may be a poor outcome measure to assess effects of training if not properly framed into more comprehensive analysis and critical appraisal of the primary studies. Later, on Limitations section you refer to those measures as very heterogenous (line 332) but it’s not possible for readers judge that unless they go and read the primary studies.
Response: We have added more detailed descriptions about measurement of CRF by incremental exercise test (line 191-199). We understand we should write up the details here to avoid readers referrning back to the primary studies. Thank you very much for the kind reminder.
– Related to my previous point I would like more detail on the results of the primary studies, e.g., how much gain was obtained in VO2max or peak in study a, b, and c. How much gain was obtained in the physical performance tests or health related quality of life measures?
Reponses: We have added more details on the results in CRF, physical functions and QoL. Thank you very much.
– I would swap the column headers and contents for the rows. I think ot would be more intuitive.
Response: That was a great suggestion. We will follow your kind advice in our future formatting the column and rows headers. We believe it would be more intuitive. Thank you very much.
Specific comments:
Line 15: Please amend “healthy” because special and clinical populations were also included in the review
Response: Thank you. We changed to healthy and clinical populations as per advice.
Line 17: Please amend “clinical trial” because not all included studies were trials, or in clinical populations.
Response: We revised as “studies” instead of “clinical trials”. Thank you very much.
Line 136: Table 3 instead of Appendix 3?
Response: Apologies for the mistake. We have replaced that with Table 3. Thank you very much.
Line 155: “risk of bias in individual studies”, what was the threshold used to decide not to conduct meta-analysis? Please report.
Response: We have added the threshold as per request and justify meta-analysis was not conducted due to the poor quality of low score (as seen in line 262-264). Thank you very much.
Line 236: “from” should be removed.
Response: Thank you. We have removed as per request.
Lines 243–247: I would like (and many other readers) details on how fatigue was assessed or the conditions of testing to critically appraise the findings.
Response: The time to fatigue described here was referred to the time to reach volitional exhaustion at the incremental test. We have restated in line 375-376 in the manuscript as per request. Thank you very much.
Reviewer 2 Report
The paper in the present version v1- is suitable for pubblication.
I suggest to the authors to report in a table the intensity and time of exercise performed in Deep Water Running, compared to studies that report similar dry land exercise, as can be obtained from the 11 studies analysed. This may improve our conclusions.
Author Response
Dear reviewer,
Thank you very much for offering us your valuable time and comments. We have revised as per your advice. Please see the attachment. Cheers!
Response to Reviewer 2 Comments
The paper in the present version v1- is suitable for publication.
Responses: Thank you very much for granting us the suitability for publication. Much appreciated. Cheers!
I suggest to the authors to report in a table the intensity and time of exercise performed in Deep Water Running, compared to studies that report similar dry land exercise, as can be obtained from the 11 studies analysed. This may improve our conclusions.
Responses: Thank you very much for your kind advice. We did refer back to the intensity of the exercise protocol performed in deep water running, however DWR studies focused on the correct techniques to perform the exercise instead of stating the intensity of the exercise. We do believe this is a valuable point and we should also address the intensity issue in our next DWR study.
And the time of exercise was included in Table 2.
Thank you very much.
Reviewer 3 Report
+ the summary reflects the content of the article
+ in the introduction, the background of the research was correctly specified
+ search sequence correctly described
+ "PRISMA flow diagram of selection process" low quality
+ Table 1-3 illegible due to orientation
+ the discussion aptly sums up the problem
+ correct conclusions
The work is very interesting and deals with an important and poorly researched problem. proposes minor editorial corrections and acceptance for publication after corrections.
Author Response
Dear reviewer,
Thank you very much for your comments and encouragement. That means a lot to us. Please see the attachment. Cheers!
Round 2
Reviewer 1 Report
Dear authors
In general, I'm satisfied with the amendments you have performed. I still find the tables very hard to read hence critically appraise the studies. If these tables were designed in a spreadsheet (e.g., Microsoft Excel) you may effortlessly switch or rotate cells (rows and columns), very easily. The TRANSPOSE function returns a vertical range of cells as a horizontal range, or vice versa (https://support.microsoft.com/en-us/office/transpose-function-ed039415-ed8a-4a81-93e9-4b6dfac76027). In addition, the order of the studies in the tables seems a little bit arbitrary, not ordered chronologically, nor by study type, nor by population... Maybe a chronologically ordered investigations would easier for readers to follow.
A few typos to amend:
line 300: delete one of the "poor"
Table 1: In the "Outcome measure of CRF" of the study "Wilbur et al., 1996", you should add a "y" to "ventilator", i.e., ventilatory.
Author Response
Dear reviewer,
Thank you very much for offering us another opportunity for revision. We took your advices on board and made changes accordingly.
The tables have been reformatted. Thank you for giving us the details in transposing. We have restructured the tables to make them clearer for readers by following chronological orders .
line 300- we deleted the "poor" as per advice. Thank you.
table 1- we revised as "ventilatory" as suggested. Thank you very much.